# Joint Selenium–Iodine Supply and Arbuscular Mycorrhizal Fungi Inoculation Affect Yield and Quality of Chickpea Seeds and Residual Biomass

**DOI:** 10.3390/plants9070804

**Published:** 2020-06-27

**Authors:** Nadezhda Golubkina, Leonardo D. Gomez, Helene Kekina, Eugenio Cozzolino, Rachael Simister, Alessio Tallarita, Valentina Torino, Andrey Koshevarov, Antonio Cuciniello, Roberto Maiello, Vincenzo Cenvinzo, Gianluca Caruso

**Affiliations:** 1Federal Scientific Center of Vegetable Production Moscow Region, 143072 Moscow, Russia; golubkina@rambler.ru (N.G.); zato@inbox.ru (A.K.); 2Center for Novel Agricultural Products, University of York, York YO10 5DD, UK; rachael.hallam@york.ac.uk; 3Department of Hygiene, Medical Postgraduate Academy, 123995 Moscow, Russia; lena.kekina@mail.ru; 4Council for Agricultural Research and Economics (CREA)—Research Center for Cereal and Industrial Crops, 81100 Caserta, Italy; eugenio.cozzolino@crea.gov.it (E.C.); antonio.cuciniello@crea.gov.it (A.C.); 5Department of Agricultural Sciences, University of Naples Federico II, Portici, 80055 Naples, Italy; lexvincentall@gmail.com (A.T.); roberto.maiello@unina.it (R.M.); vincenzo.cenvinzo2@unina.it (V.C.); 6Department of Agricultural, Environmental and Food Sciences, University of Molise, 86100 Campobasso, Italy; valentina.torino@unimol.it

**Keywords:** *Cicer arietinum* L., AMF, biofortification, proteins, antioxidants, mineral elements, waste chemical composition

## Abstract

The essentiality of selenium (Se) and iodine (I) for the human organism and the relationship between these two trace elements in mammal metabolism highlight the importance of the joint Se–I biofortification to vegetable crops in the frame of sustainable farming management. A research study was carried out in southern Italy to determine the effects of the combined inoculation with arbuscular mycorrhizal fungi (AMF) and biofortification with Se and I on plant growth, seed yield, quality, and antioxidant and elemental status, as well as residual biomass chemical composition of chickpea grown in two different planting times (14 January and 28 February). The AMF application improved the intensity of I and Se accumulation both in single and joint supply of these elements, resulting in higher seed yield and number as well as dry weight, and was also beneficial for increasing the content of antioxidants, protein, and macro- and microelements. Earlier planting time resulted in higher values of seed yield, as well as Se, I, N, P, Ca, protein, and antioxidant levels. Se and I showed a synergistic effect, stimulating the accumulation of each other in chickpea seeds. The AMF inoculation elicited a higher protein and cellulose synthesis, as well as glucose production in the residual biomass, compared to the single iodine application and the untreated control. From the present research, it can be inferred that the plant biostimulation through the soil inoculation with AMF and the biofortification with Se and I, applied singly or jointly, proved to be effective sustainable farming tools for improving the chickpea seed yield and/or quality, as well as the residual biomass chemical composition for energy production or beneficial metabolite extraction.

## 1. Introduction

I and Se are essential microelements for mammals. Se is known to be a part of the tri-iodothyronine deiodinases that participate in thyroid hormones synthesis [1]. These two trace elements are significant in antioxidant protection of the human organism; immunity maintenance; enhancement of brain activity; and protection from viral, cardiovascular, and oncological diseases [2,3]. The close relationship between the two elements entails the importance of the joint Se and I status optimization in the human organism [4]. Notably, Se and I deficiency is widespread in many countries of the world and, in this respect, plants biofortified with these two mineral elements are of great interest, providing consumption of highly available organic derivatives of Se [5] and I [6], especially considering the fact that marine products are less accessible to continental residents.

The difficulties related to this research area are associated with the scant knowledge of the relationship between Se and I in plants, where both elements are not essential [7,8], though they are able to participate in the antioxidant defense system [6,9]. Moreover, many factors affect the efficiency of biofortification, namely, the element chemical form (iodides, iodates, selenates, selenites, organic selenium, Se nanoparticles), the method of supply (soil, foliar, soilless nutrient solution), the dose, and genetic tolerance [6,10].

Previous works in this field revealed that the interaction between Se and I is not univocal—in several cases, these two elements enhanced the accumulation of each other [11], whereas in other investigations, no interaction arose [12,13,14]. Indeed, an experiment in hydroponics on spinach plants revealed no relationship between Se and I accumulation [13]. Soil application of selenite and iodate demonstrated no interaction between the two elements in biofortifying winter wheat, maize, soybean, potato, canola, and cabbage [14].

Smolen et al. found that the joint application of I and Se increased Se accumulation in lettuce leaves [12,15,16], whereas in carrot the joint application of I and Se decreased the I content compared to the single I application [17].

Moreover, the joint Se–I biofortification was investigated on Indian mustard [11], wherein plant biofortification with the single application of I or Se enhanced the leaf accumulation of Se and I by 4.3 and 2.64 times, respectively. The combined application of selenite and iodide improved Se content in pea seeds, whereas no effect of Se on I accumulation was recorded [18].

In general, the results of the previous investigations indicated a complex relationship between Se and I and a much higher efficiency of Se accumulation compared to that of I, with no effect on growth stimulation by the application of Se and I combined. In this respect, the utilization of marine algae [19,20], salicylic acid [21], and vanadium salt application [22] was employed to improve the efficiency of joint I and Se plant enrichment. However, to date, no efforts have been made to test the efficiency of arbuscular mycorrhizal fungi (AMF) inoculation. These fungi are known to establish symbiotic relationships with most plants, thus improving their nutrient and water uptake thanks to the widened root-hyphae system [23], and enhancing the plant’s resistance against biotic and abiotic stresses. Although in previous research the AMF has never been applied in combination with plant I biofortification, both alone or jointly with Se, separate investigations have reported the AMF efficiency in Se biofortification [24,25,26].

Among the targets of I–Se biofortification, plant species with intensive metabolism and high protein content are of special interest. In this respect, chickpea (*Cicer arietinum* L.) is one of the most important grain legume crops in the Mediterranean region, showing a high nutritional value [26,27] and protein level, as well as the ability to accumulate Se in the forms of SeMet and SeCys, known to be highly bioavailable for the human organism [28]. Among legumes, the chickpea is known to attain the highest Se concentration [29]. To date, no findings have been reported on I or combined I–Se biofortification in chickpea or other legumes, with this being partially due to the low I accumulation in cereal and legume seeds [30]. The exception is represented by the Se–I biofortification of pea plants [18], where iodide did not show a stimulation effect on Se accumulation upon selenate supply, contrary to selenite, for which iodide application doubled the Se concentration in pea seeds. The effect of AMF inoculation in enhancing the growth of chickpea plants was recorded by Farzaneh et al. [31].

The positive effect of AMF application on plant nutritional value may become very important in combination with Se and I biofortification. In this respect, no reports in the literature regard the biochemical characteristics of crop wastes, legumes wastes in particular, with the latter being known as a significant source of energy, fiber, protein, antioxidants, and fatty acids [32,33,34,35].

On the basis of the above-mentioned topics, the aim of the present research was to assess the interaction between Se and I under single or joint supply and AMF on seed yield, quality, and antioxidant and mineral element content, as well as on the residual biomass chemical composition of chickpea in southern Italy.

## 2. Results and Discussion

The interactions between I, Se, and AMF on yield, antioxidant status, mineral composition, protein content, and other biochemical characteristics were investigated in chickpea crops grown in two different planting times (January and February), as the experimental treatments applied are abiotic factors of great practical importance.

### 2.1. Root Mycorrhizal Colonization, Yield, Growth, and Biometrical Parameters

Indeed, the mean weight of the 1000 seeds was higher when the crop cycle started on 14 January compared to the 28 February planting, whereas the harvest index showed the opposite trend (Table 1). The latter finding is in accordance with the results of Verghis [36], who recorded significant differences in harvest index between plants grown from different sowing times. Moreover, the latter parameter attained the highest values upon the inoculation with the AMF, whereas the mean weight of the 1000 seeds was not significantly affected by joint AMF, Se, and I application. As observed in Table 1, the root mycorrhizal colonization (as a mean of the two determinations at 45 day from transplant and at the crop cycle end which did not differ statistically from each other) was not significantly affected by the crop cycle, whereas it attained the highest values in the chickpea plants inoculated with AMF in comparison with the untreated control and the biofortification treatments.

In the earlier crop cycle starting on 14 January, AMF inoculation enhanced the chickpea yield and number of seeds per plant, both without and with Se and I treatment, whereas no significant differences were recorded between the experimental treatments in the later planting time (Appendix A).

In the present research, within the selected doses of the two trace elements applied, the plant biofortification with I and Se, both singly or combined, did not significantly affect the yield parameters of chickpea seeds. In previous research, the growth stimulating effect of Se and I was recorded under the application of very narrow concentration ranges, peculiar of each plant species and cultivar. In this respect, Smolen et al. [17] demonstrated that I and Se did not affect lettuce yield. The biofortification of beans with Se revealed that seed yield largely depended on Se dose, ranging from zero up to a 1.7-fold increase [37]. A similar phenomenon has been reported for plant biofortification with I [38]. Notably, the growth stimulating effect of the combined application of Se and I has never been described to date.

The results stemming from the present investigation regarding the beneficial AMF effect on chickpea productivity are in accordance with the reported AMF growth stimulating effect on different legume species [39,40,41,42,43].

Significant differences in chickpea yield were recorded between the two planting times examined (Table 1, Appendix A), as the earlier crop cycle resulted in a longer cycle duration that enhanced the plant growth and the seed formation. With respect to the latter, the data reported in Table 2 show that the longer crop consequent to the earlier planting led to higher weights of seed dry weight (1.5 times), shoots and leaves (2.8 times), and total plant dry weight (2 times). These results are in an accordance with the findings obtained by Avelar et al. [44] in Brazil, who demonstrated that out of three sowing periods compared (May, June, July), they recorded the highest yield of chickpea seeds with the earliest planting time.

Notably, both the pods and the shoot and leaves dry weight were not significantly affected by the AMF or biofortification treatments (Table 2). Accordingly, the significant effect of AMF on the seed dry weight increase observed in the earlier crop cycle (Appendix A), reflected the corresponding enhancement of the total plant dry weight per square meter (Appendix A).

### 2.2. Quality Indicators and Antioxidants of Chickpea Seeds

As reported in Table 3, the single and combined applications of I and Se as well as the planting time did not affect the chickpea seed dry residue, which is consistent with the results of Avelar et al. [44]; only the AMF inoculation showed significant effects on the aforementioned parameters, both with or without I and Se biofotification. The increase of the total dissolved solids (TDS) was recorded under the single or joint Se, I, and AMF application, with significantly higher levels in the crop cycle starting on 28 February.

The plant inoculation with AMF mostly led to the best effects on the seed quality and antioxidant parameters when combined with the Se and I biofortification (Table 3). In particular, the single I application resulted in higher TDS, polyphenols, and antioxidant activity, whereas Se was more effective than I on the protein synthesis. In addition to its nutritional significance, this phenomenon may turn out beneficial for seed germination efficiency [45], as the antioxidants are known to protect seedlings from biotic and abiotic stresses [45]. Being an analog of S, Se reportedly affects the biosynthesis of amino acids and proteins [46] and, in this respect, an increase in rice storage protein as a result of Se biofortification was recorded by Reis et al. [47], consistently with the findings of the present investigation. In previous research [48], phenolics content and total antioxidant activity were enhanced by low doses of I, but there are no reports in the literature about the effect of the joint biofortification with Se and I concurrently with AMF inoculation on antioxidant compounds and activity, as recorded in chickpea seeds in the present study. An increase in plant antioxidant status due to AMF inoculation was previously recorded in garlic and onion fortified with Se [25]. Furthermore, AMF is known to regulate the oxidative system, hormones, and ionic equilibrium, triggering stress tolerance in plants [49]. It should be also noted that proteins, polyphenols, and antioxidant activity were better influenced by the crop cycle starting on 14 January.

### 2.3. Elemental Composition

#### 2.3.1. Selenium and Iodine Accumulation

The efficiency of I and Se accumulation in plants is governed by many factors including hormonal regulation and plant age. In this respect, as observed in Table 4, the earlier crop planting time caused the increase of Se and I content in chickpea seeds by 23% and 13%, respectively, compared to the 28 February planting time, presumably due to the longer cycle duration.

Despite the relatively low concentrations of I in chickpea seeds, I and Se showed a close relationship upon the foliar biofortification with these two elements (Figure 1). Indeed, compared to the untreated control, the plants fortified with I had a 2.6 times increase in Se accumulation in the seeds, and the combined Se + I application resulted in a twofold higher Se level than the single Se treatment (Figure 1a). The I content was enhanced by 1.4 times under the joint application of I and Se (Figure 1b). Similar Se–I relationships have been observed in previous investigations regarding the I and Se combined biofortification of Indian mustard [11].

Innovatively, the findings from the present research reveal the significant role of AMF in enhancing the I, Se, and Se + I effect on chickpea plants. This phenomenon is of special interest due to the intensive investigation of factors that can improve the combined Se + I biofortification [18,21,22]. The AMF stimulation of Se and I accumulation in chickpea seeds may be partially connected with two processes: the increase in root surface, which encourages nutrient accumulation, and the specific AMF action inside the plant. Interestingly, the AMF utilization seems to open new chances in Se/I fortification, considering that thus far only the application of phytohormones such as salicylic acid [17,21] and the vanadium compounds that participate to iodine metabolism of marine algae [22] have been investigated.

In previous investigations, the increased efficiency of Se biofortification under AMF inoculation has been documented in several plants, such as *Allium* species [25], wheat [50], and soybean [51]. Legumes have been reported to positively react to the AMF application in terms on Se accumulation, contrary to lettuce, maize, alfalfa, and forage grass (*Urochloa decumbens*) [51,52,53].

On the basis of the results of the present research, 23 grams of chickpea seeds fortified with Se upon AMF inoculation provided the adequate Se consumption level, whereas the single application of sodium selenate led to a twice lower effect.

The levels of I accumulation in chickpea seeds were rather low (Table 4), which is in agreement with the reported leaf/seed distribution of this element in plants [30,54]. With respect to the latter, the AMF beneficial effect on I biofortification recorded in the present study gives rise to high prospects for the application of these fungi in valorizing joint Se and I biofortification, even in other plant species.

#### 2.3.2. Macro-Element Content

The earlier planting time resulted in higher levels of N, P, and Ca and lower K content in chickpea seeds (Table 5); nitrates were not significantly affected by the crop cycle.

The inoculation of the AMF singly or, even more, combined with I and Se supply elicited the accumulation of Ca, Mg, P, and K in chickpea seeds compared with the untreated control (Table 5). The single I application did not show significantly higher effects than the untreated control, except for K. Notably, the biofortification with Se resulted in the highest nitrate concentration, and the combined Se + I supply also caused a higher level than the control.

Previous research investigating the effect of the single applications of Se, I, and AMF on plant elemental status confirmed the above findings, suggesting the positive contribution of AMF to the macro-element accumulation as a consequence of the nutrient uptake improvement [49]. Moreover, the modification of the macro-elemental composition elicited by the AMF inoculation to plants concurrently fortified with Se was also described in garlic, onion, and shallot [24,25].

#### 2.3.3. Microelement Content

In the present investigation, only Mn was significantly influenced by the planting time, with a higher content in the seeds grown in the later crop cycle. Most of the single and differently combined Se, I, and AMF applications enhanced the microelement concentration in the chickpea seeds compared to the untreated control, with particular reference to the treatment AMF + Se + I (Table 6), wherein Fe showed the highest content. Contrastingly, the single I supply depressed the Fe and Cu accumulation (Table 6, Figure 2), which may be connected to redox reactions between the three elements [6].

On the basis of literature reports, the AMF effect on the efficiency of Se biofortification is species-specific: in chickpea seeds, the AMF did not affect the Fe content, but increased the Zn, Cu, and Mn levels; in shallot [24], garlic, and onion [25], they elicited the Fe and Zn accumulation, but did not significantly influence Cu and Mn.

Similar to the above description, the joint biofortification with I and Se to chickpea plants increased the Zn, Fe, and Mn levels, but did not affect the Cu content in the seeds. As shown in Figure 2, both the single and combined Se, I, and AMF application caused increases of Zn in chickpea seeds, compared to the untreated control, with the highest value corresponding to the joint Se + I application followed by Se + I + AMF. No elemental composition changes were recorded in carrot by Smolen et al. [55], who deem the changes in macro- and microelement composition under both single and combined Se + I biofortification to be dependent on year rather than on Se and I.

### 2.4. Chemical Composition of Residual Biomass

Although the AMF application increased the harvest index from 40 to 45% (Table 1), about 55% of chickpea biomass remained in the field as waste. As found through the chemical analysis, cellulose, lignin, and hemicellulose were the main components of residual biomass cell walls, as they determine the stem mechanical strength. The biosynthesis of the aforementioned compounds is closely connected with each of them, despite the fact that they are remarkably different from a chemical point of view: cellulose is a glucose polymer, hemicellulose a polymer of various monosaccharides, and lignin is a high molecular weight phenylpropane polymer containing oxygen.

As observed in Table 7, the cellulose content in the cell wall residual biomass was higher upon the earlier planting time and was positively correlated with the saccharification rate, i.e., with the glucose production rate; the lignin synthesis, instead, was better enhanced by the 28 February transplant.

The AMF inoculation elicited a higher cellulose synthesis compared to the untreated control and the I biofortification, but it did not show significant improvements in comparison with the Se supply. The saccharification rate showed a positive correlation with the cellulose content and, accordingly, the same statistical differences between the experimental treatments regarding the two aforementioned variables.

The hemicellulose composition was not significantly affected by the two experimental factors applied. Xylose was the most represented monosaccharide (38.6% on average), followed by glucose (16.1%) and mannose (12.4%).

Pectin was not significantly affected by both the planting time and the AMF - biofortification treatments.

The protein content showed the same trends as the cellulose content, either referring to the crop cycles or to the AMF inoculation, and Se and I supply.

Contrary to the findings of the present investigation, in previous research carried out on broad bean [35] and sorghum [56], the authors reported a decrease in biomass lignin and hemicellulose content with the planting delay.

Significant correlations (at *p* ≤ 0.01) were recorded between the cell wall chemical compounds of the chickpea residual biomass: a negative correlation between cellulose and lignin (−0.88), cellulose and hemicellulose (−0.67), saccharification rate and lignin (−0.77), and saccharification rate and hemicellulose (−0.69), and a positive correlation between saccharification rate and cellulose (0.83), and lignin and hemicellulose (0.78).

The saccharification potential, suggesting how easily a biomass feedstock can be hydrolyzed to fermentable sugars, reflected the impact of AMF biostimulation and Se + I biofortification on the residual biomass chemical composition of chickpea plants. The high correlation coefficients between the parameters investigated express the beneficial effect of the AMF inoculation on the stem/leaf quality, similar to what was observed for the seeds. Although the biofortification with I and Se was not proven to enhance the cellulose content and, accordingly, the potential for energy production of the chickpea waste, the domestic animal feeding industry could exploit their occurrence of fiber [34]. Other possible targets associated with chickpea residual biomass utilization are represented by the baking supplements enriched with Se and I, or the extraction of Se + I containing protein additives for cosmetics [34].

The possible chances for residual biomass valorization described above exclude the goal to plough this material into the soil for supplying its content of N, P, and K to the subsequent crop. Indeed, its use as a green manure is neither practiced in wealthy countries, due to the higher use efficiency of chemical fertilizers compared to the legume-derived extra N, nor in poor agricultural areas [57]. Moreover, N from legume waste is closely dependent on the genotype [57], and it is available for the next crop absorption if there is no nitrate leaching downwards of the soil profile.

## 3. Material and Methods

### 3.1. Growth Conditions and Experimental Protocol

The research was carried out on chickpea (*Cicer arietinum* L.) landrace Cicerale, which is an ecotype originating from the National Park of Cilento and Vallo di Diano that was grown in an open field at the Department of Agricultural Sciences, University of Naples Federico II, Portici (Naples, southern Italy), in 2016–2017 and 2017–2018. The soil used for the trial was sandy-loam (76% sand, 17% silt, 7% clay), with a pH of 6.9 and an electrical conductivity of 512 mS cm^−1^; the temperature and rainfall in the two research years are shown in Figure 3.

The experimental protocol was based on the comparison between two planting times (14 January and 28 February) in factorial combination with the following eight treatments: (1) foliar supply of potassium iodide (300 mg·m^−2^, by 100 mg·L^−1^ 0.6 mM solution); (2) foliar supply of sodium selenate (150 mg·m^−2^, by 50 mg·L^−1^ 0.26 mM solution); (3) combined application of potassium iodide (KI) and sodium selenate (Na_2_SeO_4_) at the same concentrations mentioned above; (4) soil inoculation of an AMF-based formulate (Rhizotech Plus at 2 g·m^−2^ soil); (5) combined application of AMF and potassium iodide at the same concentrations mentioned above; (6) combined application of AMF and sodium selenate (Na_2_SeO_4_) at the same concentrations mentioned above; (7) combined application of AMF, potassium iodide, and sodium selenate (Na_2_SeO_4_) at the same concentrations mentioned above; and (8) untreated control. A split plot design with three replicates was used for the treatment distributions in the field, with each plot having a 4 m^2^ (2 × 2 m) surface area and 40 plants (10 plants·m^−2^).

The choice of the two planting times examined aimed to compare a 45 day transplant anticipation with the commonly chickpea crop starting on 28 February in the area where the landrace Cicerale is grown.

The four-leaf seedlings were transplanted with 20 cm spacing between the plants along the rows, which were 50 cm apart.

The AMF-based formulate Rhizotech MB (Msbiotech S.p.A., Larino, Campobasso, Italy) was applied twice, just before transplant and 45 days later; it is a plant growth-stimulating preparation that predominantly contains the endomycorrhizal fungus *Rhizophagus intraradices*, along with low concentrations of *Trichoderma harzianum* and *Bacillus subtilis*.

The foliar applications of Se and I solutions were practiced four times at 1-week intervals, starting from the first pods set.

Root mycorrhizal colonization (as a percentage) was assessed twice, 45 days after transplant and at the crop cycle end, with reference to the method of Giovannetti and Mosse [58].

The following farming practices were performed: fertilization just prior to transplant with 30 kg·ha^−1^ N, 58 P_2_O_5_, and 74 K_2_O; drip irrigation was activated when the soil available water at 10 cm depth dropped to 80%.

The chickpea pods were harvested when the plants became completely dry at the following dates: on 30 June and 7 July for the 14 January and the 28 February planting times, respectively, in 2017; on 3 and 11 July for the 14 January and the 28 February planting times, respectively, in 2018.

At harvest, the following determinations were performed in each plot: weights of seed-containing pods and of seeds, and the ratio between the two aforementioned weights; number of pods and of seeds, and their mean weight on randomly collected 200-pod samples; harvest index, calculated as the ratio between seeds and total weight per plant; dry weight of whole plants and of their parts in an oven at 70 °C until constant weight.

The residual biomass was harvested by cutting the plants at ground level and it showed no fungal symptoms; therefore, samples were randomly collected in each plot and immediately transferred to the laboratory, where they were dried in an oven at 70 °C under vacuum until they reached constant weight. After assessing the dry residue, the whole samples were carefully milled using a laboratory mill in order to avoid segregation of materials belonging to different plant organs. The final material, composed of particles <1 mm diameter, was stored in air-tight bags at −20 °C and further dried just before being processed.

### 3.2. Sample Preparation

Chickpea seeds were freeze-dried and homogenized to a fine powder and kept in polyethylene closed bags until the analysis.

### 3.3. Harvest Index

The harvest index of chickpea crops was expressed as a percentage and calculated according to the following formula: (total seeds weight): (total weight of the above ground biomass) × 100.

### 3.4. Dry Residue

The dry residue was assessed gravimetrically by drying the samples in an oven at 70 °C until constant weight.

### 3.5. Total Dissolved Solids (TDS)

TDS were determined in chickpea water extracts (1 g of dry powder in 50 mL of distilled water) using a portable conductometer TDS-3 (HM Digital Inc., Seoul, Korea). The results were calculated in milligram per gram of dry weight.

### 3.6. Nitrogen and Proteins

The total nitrogen concentration was determined according to the Kjeldahl method, and the protein content was obtained as N multiplied by 6.25 [59].

### 3.7. Polyphenols

Polyphenols were determined in ethanol extract from 1 g of dry powder of chickpea seeds by using the Folin–Ciocalteu colorimetric method as previously described [60], being expressed as milligram of gallic acid equivalents (GAE) per gram of dry weight.

### 3.8. Antioxidant Activity (AOA)

The antioxidant activity of chickpea seeds was assessed using a redox titration method via titration of 0.01 N KMnO_4_ solution with ethanolic extract [61]. The reduction of KMnO_4_ to colorless Mn^+2^ in this process reflects the quantity of antioxidants dissolvable in 70% ethanol. The values were expressed in milligram of gallic acid equivalents (GAE) per gram of dry weight.

### 3.9. Elemental Composition

Nitrates were assessed using an ion selective electrode by ionomer Expert-001 (Econix-expert, Moscow, Russia). A total of 1 g of chickpea seed powder was homogenized with 50 mL of distilled water. A quantity of 45 mL of the resulting extract was mixed with 5 mL of 0.5 M potassium sulfate background solution (necessary for regulating the ionic strength) and analyzed through the ionomer for nitrate determination.

P and Ca were determined as previously described [62]. Potassium content was detected using an ion-selective electrode. Content of Fe, Mn, Cu, and Zn in chickpea seeds was detected using an AAS spectrophotometer Shumatzu−7000 after acidic digestion of samples.

Se was analyzed using the fluorimetric method previously described for tissues and biological fluids [63]. Dried homogenized samples were digested via heating with a mixture of nitric and chloral acids, subsequent reduction of selenate (Se^+6^) to selenite (Se^+4^) with a solution of 6 N HCl, and formation of a complex between Se^+4^ and 2,3-diaminonaphtalene. Calculation of the Se concentration was achieved by recording the piazoselenol fluorescence value in hexane at 519 nm λ emission and 376 nm λ excitation. Each determination was performed in triplicate. The precision of the results was verified using a reference standard of maize grain powder in each determination with a Se concentration of 35 μg·Kg^−1^ (Agricultural Research Center, Finland).

Determination of iodine was achieved according to [64] using the Voltamperometric Analyzer TA-4 (Tomanalyte, Tomsk, Russia) equipped with built-in UV lamp and three-electrode electrochemical cell: auxiliary and reference electrodes—silver chlorides (in 1 M KCl), and working electrode—a modified silver electrode. Then, 2 mL of 10% KOH solution was added to 0.1 g of dried homogenized samples, which were ashed in a mode 40–550 °C. The mixtures were cooled down, 1 mL of 10% zinc sulfate solution was added, and ashing was repeated in the same mode. The resulting white probes were dissolved in 10 mL of distilled water, and iodine concentration was determined using concentrated formic acid as a background electrolyte and standard potassium iodide solutions of 0.1 mg·mL^−1^, 1 mg·mL^−1^, and 10 mg·mL^−1^.

### 3.10. Cellulose, Hemicellulose, Pectin, and Non-Cellulosic Monosaccharide Determinations

These determinations were performed according to the methods described by Gomez et al. [35].

### 3.11. Saccharification Assay

Loading of plant powder into 96-well plates, using a custom-made robotic platform (Labman Automation, Stokesley, North Yorkshire, United Kingdom), and saccharification assays were performed according to Gomez et al. [65] after water, acid, or alkali pretreatment. Enzymatic hydrolysis was carried out using an enzyme cocktail with a 4:1 ratio of Celluclast and Novozyme 188.

### 3.12. Statistical Analysis

Data were processed by analysis of variance, and mean separations were performed through Duncan’s multiple range test, with reference to the 0.05 probability level, using SPSS software version 21. Data expressed as percentage were subjected to angular transformation before processing.

## 4. Conclusions

From the research carried out on chickpea (*Cicer arietinum* L.) in southern Italy, it can be inferred that AMF inoculation, planting time, and a synergism between selenium and iodine have been the most important factors affecting yield, antioxidant status, and mineral and protein content, as well as Se and I accumulation in chickpea seeds.

The residual biomass left in the field at the end of the earlier crop cycle showed a better suitability towards both protein extraction and the energy production; the latter is indeed encouraged by the higher content of cellulose and the correspondent higher saccharification rate.

The combined strategy including the AMF inoculation along with the Se and I supply proved to represent a new approach for enhancing the yield and/or the quality and nutritional value of seeds and crop waste.

## Figures and Tables

**Figure 1 plants-09-00804-f001:**
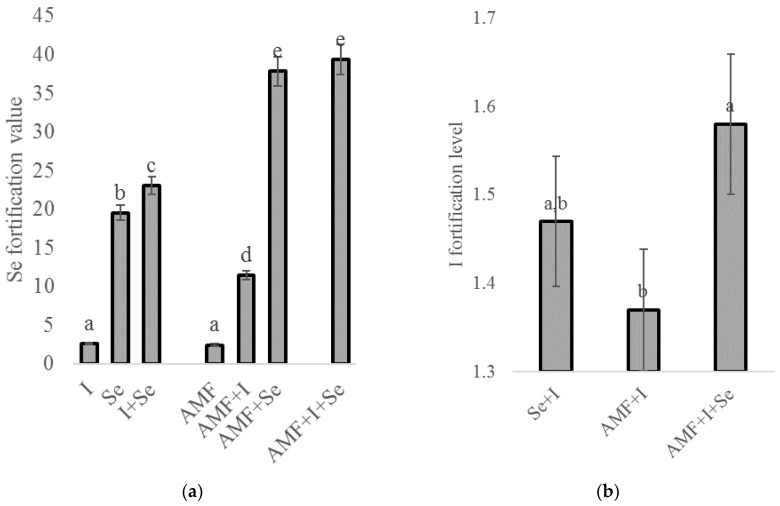
Effect of AMF, Se, and I biofortification on Se (**a**) and I (**b**) biofortification values. Values followed by the same letter are not significantly different according to Duncan’s test at *p* < 0.05.

**Figure 2 plants-09-00804-f002:**
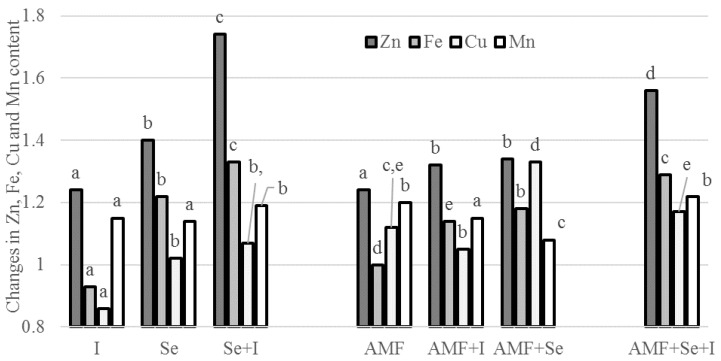
Effect of AMF, Se, and I application on microelement content in chickpea seeds. Values followed by different letters are significantly different according to Duncan’s test at *p* ≤ 0.05.

**Figure 3 plants-09-00804-f003:**
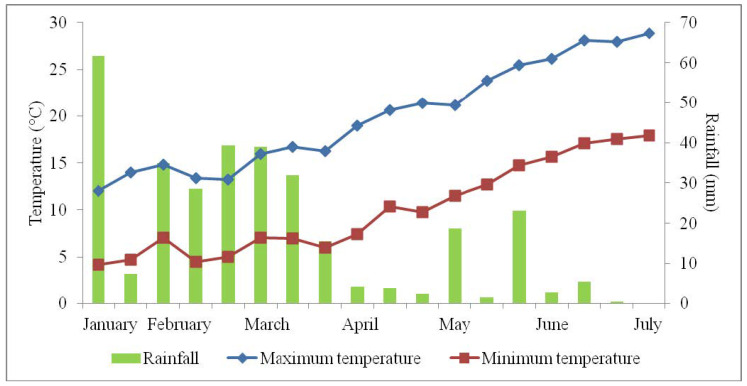
Trend of temperature and rainfall in Portici (Naples, Italy): mean values of 2016–2017 and 2017–2018, from the second 10-day interval of January to the first 10-day interval of July.

**Table 1 plants-09-00804-t001:** Yield parameters as affected by planting time and arbuscular mycorrhizal fungi (AMF) - biofortification.

	Root Mycorrhizal		Seed		Seeds		1000 Seeds	Seed/	Harvest	
Experimental Treatment	Colonization		Yield		No. Per Plant		Weight	Pod Weight	Index	
	%		t·ha^−^^1^				g	%	%	
Planting time										
14 January	42		6.44		134.1		480.3	75.1	37.4	
28 February	44		4.29		111.7		384.7	74.6	48.4	
	n.s.		*		*		*	n.s.	*	
AMF - Biofortification										
Control	31	b	5.02	b	117.8	b	424.3	74.7	40.5	b
Se	33	b	5.08	b	116.6	b	433.3	74.8	41.6	b
I	32	b	5.03	b	118.9	b	420.3	74.6	40.5	b
Se + I	33	b	5.12	b	119.2	b	425.4	74.1	40.9	b
AMF	53	a	5.59	a	126.3	a	437.7	74.4	44.5	a
AMF + Se	54	a	5.71	a	126.2	a	446.7	75.2	45.0	a
AMF + I	53	a	5.56	a	126.6	a	433.5	75.1	44.5	a
AMF + Se + I	55	a	5.80	a	131.7	a	438.9	76.1	45.5	a
							n.s.	n.s.		

n.s.: not statistically significant; * significant at *p* ≤ 0.05. Within each column, values followed by different letters are significantly different according to Duncan’s test at *p* ≤ 0.05.

**Table 2 plants-09-00804-t002:** Biometrical and growth parameters as affected by planting time and AMF - biofortification.

Experimental Treatment	Pods No. per Plant	Seeds No. per Pod		Seeds Dry Weight g per m^2^		Pods Dry Weight g per m^2^	Shoots + Leaves Dry Weight g per m^2^	Plant Dry Weight g per m^2^	
Planting time									
14 January	111.6	1.20		575.0		190.6	785.1	1550.7	
28 February	92.3	1.21		382.8		130.1	278.4	791.3	
	*	n.s.		*		*	*	*	
AMF - Biofortification									
Control	103.1	1.15	b	448.3	b	153.0	529.6	1130.8	b
Se	101.2	1.15	b	453.8	b	151.3	524.3	1129.4	b
I	102.4	1.16	b	449.3	b	152.8	527.7	1129.8	b
Se + I	102.0	1.17	b	457.0	b	160.1	521.7	1138.7	b
AMF	100.9	1.25	a	499.1	a	170.7	539.0	1208.9	a
AMF + Se	100.9	1.25	a	509.7	a	166.9	534.6	1211.2	a
AMF + I	101.3	1.25	a	496.3	a	165.2	540.7	1202.3	a
AMF + Se + I	103.9	1.27	a	517.8	a	162.7	536.9	1217.3	a
	n.s.					n.s.	n.s.		

n.s.: not statistically significant; * significant at *p* ≤ 0.05. Within each column, values followed by different letters are significantly different according to Duncan’s test at *p* ≤ 0.05.

**Table 3 plants-09-00804-t003:** Quality indicators and antioxidants of chickpea seeds as affected by planting time, AMF inoculation, and Se and I biofortification.

Experimental Treatment	Dry Residue %	TDS mg·g^−1^ d.w.	Proteins g·kg^−1^ d.w.	Polyphenols mg GAE·g^−1^ d.w.	Antioxidant Activity mg GAE·g^−1^ d.w.
Planting time					
14 January	88.8	14.7	170.2	10.5	21.1
28 February	89.1	15.2	141.9	9.4	18.6
	n.s.	*	*	*	*
AMF - Biofortification					
Control	87.8 b	13.5 e	128.2 e	8.8 c	17.8 e
Se	88.5 ab	14.1 d	152.0 cd	9.1 c	17.8 e
I	88.0 b	15.2 c	128.2 e	10.1 ab	19.1 d
Se + I	88.6 ab	15.3 bc	149.9 d	9.9 b	20.1 cd
AMF	89.0 a	14.5 d	164.3 b	10.2 ab	20.7 bc
AMF + Se	89.3 a	15.7 ab	178.4 a	10.5 a	21.9 a
AMF + I	89.1 a	15.6 ac	159.9 bc	10.5 a	20.0 cd
AMF + Se + I	89.5 a	15.8 a	187.3 a	10.5 a	21.4 ab

d.w.: dry weight; n.s.: not statistically significant; * significant at *p ≤* 0.05. Within each column, values followed by different letters are statistically different according to Duncan’s test at *p*
*≤* 0.05. TDS: total dissolved solids.

**Table 4 plants-09-00804-t004:** Se and I content in chickpea seeds, as affected by planting time and AMF - biofortification.

Treatment	Se	I
	µg·kg^−1^ d.w.	
Planting time		
14 January	1632.5	13.8
28 February	1252.5	12.0
	*	*
AMF - Biofortification		
Control	84.0 f	0.0 d
Se	1642.5 c	0.0 d
I	219.5 e	9.5 c
Se + I	1937.5 b	13.0 b
AMF	207.0 e	0.0 d
AMF + Se	3177.0 a	0.0 d
AMF + I	967.5 d	14.0 ab
AMF + Se + I	3305.0 a	15.0 a

d.w.: dry weight; * significant at *p* ≤ 0.05. Within each column, values followed by different letters are statistically different according to Duncan’s test at *p*
*≤* 0.05.

**Table 5 plants-09-00804-t005:** Macro-element contents in chickpea seeds as affected by planting time and AMF - biofortification.

Treatment	N	P	K	Ca	NO_3_
	g·kg^−1^ d.w.				mg·kg^−1^ d.w.
Planting time					
14 January	27.2	3.21	8.86	1.39	589.1
28 February	22.7	2.67	9.43	1.14	578.6
	*	*	*	*	n.s.
AMF - Biofortification					
Control	20.5 e	2.36 d	8.64 d	1.00 d	550.0 cd
Se	24.3 cd	2.80 c	8.83 cd	1.22 c	673.0 a
I	20.5 e	2.40 d	9.21 ac	1.03 d	588.0 bc
Se + I	24.0 d	2.87 c	9.29 ab	1.23 c	625.5 b
AMF	26.3 b	3.08 b	9.24 ab	1.34 b	594.0 bc
AMF + Se	28.5 a	3.40 a	9.58 a	1.46 a	567.5 c
AMF + I	25.6 bc	3.13 b	9.06 b	1.34 b	545.0 cd
AMF + Se + I	30.0 a	3.48 a	9.29 ab	1.50 a	528.0 d

d.w.: dry weight; n.s.: not statistically significant; * significant at p ≤ 0.05. Within each column, values followed by different letters are statistically different according to Duncan’s test at p to Duncan’s test at *p*
*≤* 0.05.

**Table 6 plants-09-00804-t006:** Microelement content in chickpea seeds as affected by planting time and AMF - biofortification.

Treatment	Zn	Fe	Cu	Mn
	mg·kg^−1^ d.w.			
Planting time				
14 January	8.5	28.1	4.5	8.2
28 February	8.3	28.7	4.6	8.7
	n.s.	n.s.	n.s.	*
AMF-Biofortification				
Control	6.2 e	25.0 d	4.2 d	7.4 d
Se	8.7 c	30.5 b	4.3 d	8.4 b
I	7.7 d	23.3 e	3.6 e	8.5 b
Se + I	10.8 a	33.3 a	4.5 cd	8.8 a
AMF	7.7 d	24.9 d	4.7 bc	8.9 a
AMF + Se	8.3 c	29.6 b	5.6 a	8.0 c
AMF + I	8.2 c	28.4 c	4.4 d	8.5 b
AMF + Se + I	9.7 b	32.3 a	4.9 b	9.0 a

d.w.: dry weight; n.s.: not statistically significant; * significant at *p* ≤ 0.05. Within each column, values followed by different letters are statistically different according to Duncan’s test at *p* ≤ 0.05.

**Table 7 plants-09-00804-t007:** Chemical composition of chickpea residual biomass as affected by planting time and AMF - biofortification.

Treatment	Lignin %		Cellulose %		Hemicellulose %		Pectin %	Proteins g·kg^−1^		Saccharification Rate g Glucose kg^−1^ h^−1^	
Planting time											
14 January	15.4		44.4		14.1		6.3	120.6		30.8	
28 February	18.1		40.0		14.7		6.6	100.4		27.4	
	*		*		n.s.		n.s.	*		*	
AMF - Biofortification											
Control	18.1	a	39.6	b	15.7	a	6.2	91.0	e	27.3	b
Se	17.0	ab	41.3	ab	15.0	ab	6.5	107.7	e	28.5	ab
I	18.0	a	39.8	b	15.7	a	6.3	91.3	cd	27.4	b
Se + I	16.8	ab	41.5	ab	14.8	ab	6.5	105.5	d	28.7	ab
AMF	16.3	b	43.5	a	13.9	bc	6.6	116.3	b	30.0	a
AMF + Se	16.1	b	44.3	a	13.4	c	6.7	127.2	a	30.4	a
AMF + I	16.3	b	43.6	a	13.8	bc	6.5	113.1	bc	29.9	a
AMF + I + Se	16.0	b	44.2	a	13.3	c	6.6	132.0	a	30.5	a
							n.s.				

All the values are expressed as dry weight basis; n.s.: not statistically significant; * significant at *p* ≤ 0.05. Within each column, values followed by different letters are statistically different according to Duncan’s test at *p* ≤ 0.05.

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
