# Peer review of "Joint Selenium–Iodine Supply and Arbuscular Mycorrhizal Fungi Inoculation Affect Yield and Quality of Chickpea Seeds and Residual Biomass"

_plants, 2020, doi:10.3390/plants9070804_

Round 1

Reviewer 1 Report

Title: Joint Selenium-Iodine Supply and Arbuscular Mycorrhizal Fungi Inoculation Affect Yield and Quality of Chickpea Seeds and Residual Biomass. Golubkina N, Gomez LD, Kekina H, et al.

This work describes the improvement of the chickpea seed yield and quality using arbuscular mycorrhizal fungi inoculation and selenium and iodine supply and studies the residual biomass left in the field at the end of the crop cycles.

I found the paper original and of a good quality, and that can enlarge the scientific knowledge about the interesting and important topic of plant biostimulation and biofortification. Nevertheless, I found some mistakes in the redaction of the manuscript as well as in the display and description of the results. Therefore, I recommend a minor revision due to the experiments are well performed and I consider that the work is complete, although some of the requested changes in the manuscript are not minor changes. I ask that the authors specifically address each of my comments in their response.

Comments:

Please, review the redaction of the complete manuscript. Check the enumerations, for example line 65 and line 285 (“…that Se adequate consumption level…”). Some conjunction should be used at the end of the enumerations, like “and”. Pay attention to the English writing, there are some errors (example: line 268).

Abbreviations: please, review them and indicate only the first time that they appear, using parentheses (example: line 52, “Iodine (I) and selenium (Se)…”). After that, use the same abbreviation when the concept is used (example: modify lines 43 and 83).

Abstract: it is too long. Please, shorten it; it is not necessary to be very specific with the results obtained.

Tables: please, include the standard deviations or errors when possible. Describe better the tables in the captions. Some parameters like the harvest index should be explained in the captions and/or in the results redaction.

Figures: Please, include error bars when possible (example: in figure 2).

Introduction: Please, explain why you use arbuscular mycorrhizal fungi and describe something about them. Lines 104-107: please, explain the reason of the chosen dates for the experiments.

Results and discussion: the authors describe directly the results observed in the tables or figures. Please, make a small introduction explaining the aim of every experiment and what is already known about that.

Review well the description of the results, there are some mistakes (example: line 227 and 304).

Section 2.3 and 2.3.1: It is hard to understand the description of the results according to the table. Please, make a division in the table or split it in two tables. Please, indicate figure 5a or 5b to make the results easier to understand.

Lines 259-264: it is the same paragraph as the previous one (lines 246-250).

Section 2.4 is missing and figure 6 is not described in the manuscript.

Some results are a little redundant in some tables and figures. As an example, table 5 and figures 5 and 6. Please, include in these cases the table as supplementary material.

Lines 324-328: please, refer it to a figure, table or results not shown, and explain it more clearly. It is difficult to understand the paragraph.

Discussion: I consider that the discussion is not very well structured. Some sections have a faint discussion and other a more extended. Please, review that the discussion of the different sections is homogeneous.

Author Response

Dear Reviewer, we have uploaded the answers to your comments. Thanks for your contribution to improve our manuscript!

Reviewer 2 Report

In the manuscript entitled "Joint Selenium-Iodine Supply and Arbuscular Mycorrhizal Fungi Inoculation Affect Yield and Quality of Chickpea Seeds and Residual Biomass" the research was carried out to determine the effects of the combined inoculation with AMF and biofortification with Se and I on plant growth, yield of seeds, quality of antioxidants and status of yield enrichment with some elements. Additionally, the chemical composition of residual biomass of plants grown in two different planting times was verified. Authors informed us that AMF application to Cicer arietinum culture  planted into the field 14 January, that is in the earlier time of planting, has resulted in higher seed yield, the number of seeds, and plant dry weight. This planting time  proved better in terms of N, P, Ca, Se and I contents in checked seed samples. In both crop cycles the  AMF-treatment and the biofortification with selenium and iodine positively impacted the level of antioxidants and content of nutrients in cultivated chickpea  . The residual biomass  proved to be suitable in terms of possible protein extraction or the energy production in the frame of sustainable farming management. The findings from the presented research reveal the significant role of AMF in the effectiveness of iodine, Se and Se I biofortification in chickpea, and in particular the AMF ability to  improve iodine accumulation. The text is interestingly written and the results obtained are valuable, but I think that the manuscript should be sorted out a bit, adapting it to the requirements of the magazine to which it was submitted. I suggest keeping the classic order of the subsections because in the current situation we see the Conclusions after M&M, which in turn we may read only after reading the results. By the way, I ask the authors to think again about the graphic presentation of the results so that they can be better understood by the reader. And a research hypothesis would be appreciated.

Author Response

(The authors gave the same response as above.)
